# Barriers and facilitators to social inclusion among people with severe mental illness: A qualitative study

Sharon Eager[1]*, Brynmor Lloyd-Evans[1], Jennifer Bousfield[2], Joanna C.[1], Megan Downey[3], Isobel Harrison[1], Helen Killaspy[1,3], Gillian Mezey[2]

1 Division of Psychiatry, University College London, London, United Kingdom, 2 Population Health Research Institute, St George's University of London, London, United Kingdom, 3 Camden and Islington NHS Foundation Trust, London, United Kingdom

* sharon.eager.20@ucl.ac.uk

## Abstract

People with severe mental illness are often socially excluded. Social exclusion is associated with worse mental health outcomes and poorer quality of life, making it a key target for recovery. Despite this, people with severe mental illness often do not receive support for social inclusion. Further, it has not been clearly established how to improve social inclusion in this population. This study aimed to qualitatively examine the barriers and facilitators to social inclusion among people with severe mental illness, from the perspective of both mental health service users and staff. We conducted individual interviews with people with a severe mental illness and focus groups with mental health staff. Both service user and staff participants were recruited from a range of mental health services in two NHS trusts in London. Questions explored participant views on the barriers and facilitators to social inclusion among people with severe mental illness, using semi-structured topic guides. Interviews and focus groups were analysed using inductive thematic analysis. Interviews were conducted with fifteen service users and four focus groups were conducted with sixteen mental health staff. We identified six primary themes representing a range of barriers and facilitators to social inclusion. These related to: (i) a person's own individual characteristics and abilities; (ii) the role of their mental illness; (iii) the influence of their social relationships; (iv) the system of mental health care and support; (v) their financial situation and access to opportunities; and (vi) the impact of different types of stigma. Social inclusion is a complex but important priority for people with severe mental illness. This study highlights the particular need for continued development into social interventions, an individualised approach to care, and stigma reduction in mental health.

**Data availability statement:** In compliance with our ethical approval from the London-Bromley Research and Ethics Committee (approval ID: IS/LO/1778), individual transcripts from this study cannot be publicly shared. Participants did not consent to the disclosure of these transcripts, and they contain sensitive and potentially identifying information. Therefore, public access would compromise patient confidentiality. All relevant data are included within the manuscript. Data access requests can be directed to the corresponding author (sharon.eager.20@ucl.ac.uk) or the London-Bromley Research and Ethics Committee (bromley.rec@hra.nhs.uk).

**Funding:** The paper reports work funded by the National Institute for Health and Care Research, School for Social Care Research (NIHR SSCR). The grant reference is C088/T14-035/SGGM-P68 (GM, HK, BLE, JB, and IH). The funders had no role in study design, data collection and analysis, decision to publish, or preparation of the manuscript.

**Competing interests:** The authors have declared that no competing interests exist.

## Introduction

Social exclusion refers to an individual's lack of opportunity to experience the typical rights, relationships, and activities available to most people in society, typically for reasons that are beyond their control [1,2]. Social inclusion, meanwhile, is defined as having access to social and economic opportunities, and a sense of community belonging [2,3]. It is a multi-dimensional and dynamic concept [3], that has been conceptualised to encompass five key domains: productivity (including education and employment), social integration (including social relationships and networks), consumption (including poverty and financial wellbeing), access to services (including primary care), and political engagement (including voting and campaigning) [4].

Severe mental illness refers to a psychiatric diagnosis that results in considerable role impairment in one or more life domains (e.g., social relationships, employment, independent living) and an extended period of time receiving support from mental health services [5,6]. People with severe mental illness are among the most socially excluded individuals in society [7]. From the perspective of key mental health stakeholders, including people with lived experience of serious mental illness, the key social inclusion goals for this population include participation in social activities, housing, community involvement, employment, health, and service utilisation [8]. Further research demonstrates that people with serious mental health problems often have low levels of employment [9,10], smaller and less satisfying social networks [11], poor social recovery [12], and suffer high levels of criminal victimisation [13,14]. Other key unmet social inclusion needs for such individuals include loneliness, financial support, and romantic relationships [15].

Furthermore, social exclusion is thought to both contribute to and be a consequence of mental ill-health [16]. Some aspects of social inclusion, namely productivity and social integration, have been found to reduce after mental illness onset [17]. For this population, social exclusion is associated with worse mental health outcomes, more severe psychiatric symptoms, and poorer quality of life [18,19]. Such findings suggest that social inclusion is a key area for mental health practitioners to try to address [19,20], however, those with serious mental illness often do not receive targeted support for social inclusion [21,22]. Additionally, there has been relatively little empirical research focused on understanding how best to help with social exclusion in a mental health context [23]. Therefore, it is first important to identify the factors that act as barriers to social inclusion, and those that facilitate it, to assess how best to improve social inclusion in this population [24].

Several possible barriers to social inclusion in this population have been proposed, including the nature of mental health symptoms themselves, poor self-esteem, low confidence, relationships, discrimination, and lack of opportunity [16,17]. These factors operate across micro (individual), meso (groups and relationships), and macro (societal) levels of influence [2,25,26]. However, there is a lack of in-depth research examining the barriers and facilitators to social inclusion according to people with lived experience of mental health problems and mental health professionals [8]. Such insight can help inform the development of relevant and

targeted support for social exclusion and promote understanding of key aspects of recovery for service users themselves [27]. These principles are in line with recovery-oriented practice [28]. The current study aimed to qualitative examine the barriers and facilitators to social inclusion in people with severe mental illness, from the perspectives of those with lived experience and mental health professionals.

## Methods

### Study design

We conducted a qualitative study to explore service user and staff views on the barriers and facilitators to social inclusion among people with severe mental illness. We carried out in-depth, individual interviews with service users to enable participants to speak openly about their experiences. We conducted focus groups with staff to encourage varied perspectives and collaborative discussion. We used semi-structured topic guides for the interviews and focus groups. Ethical approval was granted by London-Bromley Research and Ethics Committee (IS/LO/1778).

### Sample and recruitment

Service user participants represented a subgroup of individuals participating in a larger, earlier study to develop a measure of social inclusion – the SInQUE (Social Inclusion Questionnaire User Experience) [4]. This larger study sample comprised 192 adult users of secondary mental health services in London with a primary diagnosis of a psychotic illness, common mental disorder, or personality disorder. These participants were recruited from 22 different mental health teams across two NHS trusts: South West London and St George's Mental Health NHS Trust and Camden and Islington NHS Foundation Trust. Participating teams included community mental health teams, community rehabilitation teams, complex depression and trauma teams, personality disorder services, assertive outreach teams, early intervention for psychosis services, and community forensic services. Further details of recruitment for the larger study are reported elsewhere [4,19]. Most participants (165 of 192) consented to be contacted for further qualitative interviews; 27 were either unable to be contacted or declined to take part. Of these, 138 agreed to take part in an in-depth interview to explore their experience and views of social inclusion. From these, 15 participants were purposively recruited for individual interviews, to include people of different gender, age, ethnicity, and diagnosis.

Participants for staff focus groups were also recruited from those involved in the larger study that developed the SInQUE [4]. Staff from the 22 participating mental health teams were sent an email inviting them to participate in a focus group. Four focus groups were conducted (two in each participating NHS trust) with staff purposively recruited to include people from a range of teams and with varying roles and experience.

### Topic guide

Semi-structured topic guides were developed by members of the research team, informed by their previous work on this topic. The topic guide for the service user interviews opened with general questions about the participant's mental health, their living situation, their work life, and their family situation. A brief explanation of the study and a definition of social inclusion were then provided, prior to the main interview questions. Questions and prompts related to situations where participants had felt excluded, their current support, the effect of their mental illness on their experiences of social inclusion, and anything that had helped them feel more socially included.

As well as specific questions about the SInQUE questionnaire, the topic guide for the staff focus groups included broader prompts relating to domains of social inclusion. These included their perceptions of how to improve service user social inclusion, barriers to this, and any strategies they use to promote social inclusion amongst service users. The topic guides were reviewed by two service user research advisory groups and a stakeholder advisory group from the larger SInQUE study and were refined according to their feedback. The topic guides are included in S1 File.

### Data collection

Potential service user participants were contacted and invited to take part in an interview either in their own home or in their community mental health team centre, according to their own preference. Staff were emailed an invitation to attend one of two focus groups that had been arranged in community mental health centres within their respective NHS trust. All interviews and focus groups were conducted in-person by two members of the research team (JB and IH). Written informed consent was obtained at the start of each interview and focus group. During the consent procedure, we asked service users for permission to inform, and if necessary, obtain support from their GP or care coordinator if a participant were to become distressed or if there were immediate safety concerns. During the interviews, although a few participants mentioned previous suicidal ideation in the context of social exclusion, there were no present serious safety concerns requiring researchers to follow the safety protocol. All service user interviews were conducted between 20th October 2016 and 21st April 2017. Staff focus groups were conducted between 4th April 2017 and 18th May 2017. Interviews and focus groups lasted for approximately one hour and were audio recorded. Service user participants were offered £20 in recognition of their time.

### Data analysis

Audio recordings were transcribed by an independent transcription service and transcripts were checked for accuracy by the researchers. Transcripts were anonymised by removing any potentially identifiable information. Using the analytic software NVivo 12, interviews and focus groups were analysed using reflexive thematic analysis [29]. Analysis was conducted collaboratively among the research team. SE initially coded four interviews and one focus group and developed a coding framework with preliminary themes based on this. This preliminary framework was shared with five members of the research team along with one different interview/focus group transcript each. This collaborative approach enabled the integration of diverse academic, clinical, and lived experience perspectives in the analytic process from researchers with backgrounds in social work (BLE), rehabilitation psychiatry (HK), forensic psychiatry (GM), community mental health rehabilitation (MD), and lived experience of mental health service use (JC).

Each individual independently coded their respective interview/focus group transcript using the preliminary framework and noted their suggested modifications to the coding and thematic framework initially developed by SE. The members of the team involved in coding met to discuss these proposed changes and SE adjusted the coding framework and themes, integrating their feedback. This modified framework was distributed to the same team members, with a different transcript to analyse. Again, the team independently analysed one interview/focus group transcript each using the modified framework and suggested further minor alterations. At this stage, a high level of agreement on the suitability of the coding and thematic framework was observed across all members of the team. SE integrated these suggestions into a finalised framework, which the team reviewed and approved. SE then used this coding framework to analyse all 15 interviews and 4 staff focus groups.

## Results

### Participants

15 service users participated in individual interviews and 16 staff members participated across four focus groups (with six staff members in one group, four in another, and two focus groups of three staff each). Service user participants were aged between 26 and 58 years old and had a mean age of 37.4 years (SD = 9.2). Most were of white ethnicity and the majority had a primary diagnosis of a psychotic disorder (Table 1). Staff participants worked across a range of professional mental health roles (Table 2).

### Themes

We identified six primary themes, each of which encompassed further related subthemes (Table 3). These themes capture service user and staff participant experiences of the barriers and facilitators to social inclusion among people with severe

**Table 1. Service user participant demographic characteristics.**

| Characteristics | Participants (N = 15) |
|---|---|
| **Mean age (SD)** | 37.4 (9.2) |
| **Gender** | |
| Male | 8 (53.3%) |
| Female | 7 (46.7%) |
| **Ethnicity** | |
| White (White British/White Irish/White European) | 11 (73.3%) |
| Black (Black British/Black Caribbean/Black African) | 3 (20.0%) |
| Asian (Asian British/Indian/Pakistani/Bangladeshi/Chinese) | 1 (6.7%) |
| **Diagnostic group** | |
| Psychosis (schizophrenia, schizoaffective disorder, bipolar disorder, or first-episode psychosis) | 10 (66.7%) |
| Common mental health disorder (depression, anxiety, obsessive compulsive disorder, or post-traumatic stress disorder) | 2 (13.3%) |
| Personality disorder | 3 (20.0%) |

Data presented in the format N (%), unless otherwise stated.

N = number of participants.

SD = standard deviation.

**Table 2. Staff focus group job roles.**

| Job role | Participants (N = 16 across 4 focus groups) |
|---|---|
| Occupational therapist | 4 (25.0%) |
| Care coordinator | 1 (6.3%) |
| Psychiatrist | 2 (12.5%) |
| Nurse | 5 (31.3%) |
| Social worker | 2 (12.5%) |
| Assistant psychologist | 1 (6.3%) |
| Team manager | 1 (6.3%) |

Data presented in the format N (%).

N = number of participants.

mental illness, organised according to their level of impact (micro, meso, or macro). Many of these factors were possible to conceptualise as either barriers or facilitators to social inclusion, depending on whether they were present or absent. Therefore, we have not separated these into distinct barriers and facilitators. Generally, there was a high level of congruence between the views of service user and staff participants. We have noted in the text any subthemes that were exclusively mentioned by either service user or staff participants.

## Individual characteristics and abilities

Service user participants reflected on various ways their individual characteristics and abilities acted as barriers and facilitators to social inclusion. Many felt that low self-confidence and lack of self-esteem were key barriers to their social and societal engagement:

**Table 3.** Themes and subthemes reflecting barriers and facilitators to social inclusion.

| Level of impact | Theme | Subthemes |
| --- | --- | --- |
| Micro | Individual characteristics and abilities | • Low confidence and lack of self-esteem<br>• Difficulty with social and language skills<br>• Age<br>• Self-initiative and proactivity |
| | Role of mental illness | • Disabling mental health symptoms<br>• Social withdrawal as a coping mechanism<br>• Positives and negatives of medication and diagnosis |
| Meso | Social relationships | • Role and proximity of family members<br>• Role and proximity of friends<br>• Having opportunity to meet people<br>• Trust/mistrust and vulnerability from others |
| | System of mental healthcare and support | • Relationship with staff<br>• Knowledge of staff about local resources<br>• Availability of relevant support and priorities of system |
| Macro | Finances and access to opportunities | • Poverty and financial benefits/support<br>• Access to employment<br>• Access to education |
| | Stigma | • Internalised, self-stigma<br>• Stigma from family, friends, and romantic partners<br>• Stigma in mental health services<br>• Stigma in education and workplace<br>• Societal stigma |

*"I think my self-confidence has been quite low and obviously that's because of the experiences that I have been through. So yeah, I think my confidence, my self-confidence is quite low. And I think that affects everything."*

(Service user participant 10, 35-year-old female)

Some service user participants described how feelings of insecurity hindered their pursuit of personal relationships, including friendships and romance. Some felt they lacked social skills and struggled to relate to other people. One noted their lack of computing skills as a particular challenge:

*"Well understanding the computer; how to use technology, that can be… I find that challenging. Other people… duck to water but, you know, I find that very challenging."*

(Service user participant 12, 53-year-old female)

Staff participants also highlighted how language skills can be an important factor to consider in relation to service users' social inclusion:

*"I think also in CAMHS [Child and Adolescent Mental Health Services] we've got quite a high percentage of our patients who don't speak English as a first language; that can be a barrier to social inclusion as well."*

(Staff focus group 1)

Both service user and staff participants also talked about the role that age can play in social inclusion. Some interviewees felt that getting older made meeting people and making friends more difficult. Sometimes, older age presented a barrier to finding work or engaging in training. Others, however, felt that getting older and feeling that they had missed out on opportunities due to their mental health had motivated them to engage in activities and pursue their interests:

*"I think there's a sense of urgency because so many years have been…I mean alright I'm still young, but a lot of years have been wasted because of mental illness. So there is a sense of urgency to get on with volunteering and all of that – and all the things I wanna do, you know?"*

(Service user participant 14, 27-year-old male)

Some service user participants highlighted a desire to use their own experiences to help other people as a source of motivation to seek out relevant opportunities. They discussed not wanting their negative experiences to 'go to waste' and felt that they could offer hope and motivation to other people who may be who may be struggling with similar problems to their own.

Several service users emphasised self-initiation and proactivity as key facilitators for social inclusion. It was felt that social integration was more likely to be effective if the desire for inclusion came from the individual themselves, rather than being imposed on them:

*"If I make the decision myself I think I get more ownership of the activity. […] Whereas if people are pushing me to do something then I have a tendency to recoil or rebel against that."*

(Service user participant 6, 31-year-old male)

This view was echoed by staff participants. They felt that it was important not to make assumptions about what someone might want or need, but to instead help the person to identify their priorities through discussion:

*"I think it kind of touches on how people define social inclusion for themselves and how excluded they feel. […] So you know, if they want to return to work, but others might not, and that might not be a priority. So to kind of force that on somebody is a bit of our goal, not their goal."*

(Staff focus group 2)

### Role of mental illness

Service user participants described ways that their mental illness affected their social inclusion. Frequently reported were mental health symptoms, including hallucinations, paranoia, depression, anxiety, and suicidality, which limited and restricted their personal relationships and wider societal involvement.

*"There's been many occasions where there's been an opportunity for me to say something suggestive or to actually kiss a girl, but the voices have attacked me and stopped me from doing it – just at that crucial point, you know?"*

(Service user participant 13, 36-year-old male)

One cited suicidal ideation as a reason for not thinking about life practicalities and planning beyond the short-term:

*"If I don't wanna live then I don't have to worry about 'what shall I do next year? Do I wanna be an accountant? How is my mum?' You know, having a relationship, having family, children, old age, jobs, commuting – you don't have to worry about mundane things."*

(Service user participant 1, 35-year-old male)

Staff participants also identified that poor self-care and hygiene, which may be a problem for some individuals with severe mental illness, can result in societal and interpersonal rejection and be a barrier to social inclusion:

*"I've noticed people that are malodorous or have poor hygiene, I think that's really difficult to socially integrate you know with groups – even if they're willing to go, they kind of get avoided a bit."*

(Staff focus group 1)

Some service user participants talked about social withdrawal as a symptom of their mental illness and the negative impact this had on their social inclusion:

*"If I'm by myself and not really seeing people, or when I've been unwell, I tend to withdraw from everybody and just kind of hibernate in my life."*

(Participant 10, 35-year-old female)

Psychotropic medication was referred to as both a barrier and a facilitator of social inclusion. Some service user participants complained about the negative side effects that came with their medication, including weight gain and fatigue, which negatively impacted their social life. Others, however, acknowledged the positive affect that medication had on their lives and their ability to engage with others. Some felt that establishing the right medication was a key step in helping them feel more stable and in control of their lives, and thus better able to pursue work:

*"I was always changing medicine and I wasn't very well at all. […] Then I think when I was 21, 22, they put me on Clozapine and I started to have it stronger and I started working. So then when I started working it gradually… it got better."*

(Service user participant 8, 36-year-old female)

Being given a diagnosis also helped service users contextualise their difficulties with social inclusion and provide a solution, rather than attributing difficulties to personal weakness or inadequacy:

*"I didn't have a diagnosis until I was 20 or 21, I think. So it was actually a relief to get my diagnosis, because I had something to call it; you know, and I could read up on it, and I could see how I could have medicine to make me better and all of that sort of thing. Before I had nothing, and I just didn't know what was wrong with me."*

(Service user participant 3, 38-year-old female)

## Social relationships

Service user participants discussed the key role that family members played in their lives, and how family could act as both a barrier and facilitator for social inclusion. Most highlighted the positive influence family had, indicating that family members could provide direct support and encouragement, but could also hold them accountable when needed. One described how their family was a key source of encouragement and support, both directly in relation to helping them manage their mental health problems and more widely with social engagement:

*"I couldn't cope without their help. […] They encourage me, and they give me company, and they do stuff with me, and my mum helps me with forms or whatever. They take me to places, they phone me up and they're always encouraging and interested in me, suggesting ideas the whole time."*

(Service user participant 13, 36-year-old male)

However, some felt that their family was a source of stress that made certain facets of social inclusion more difficult for them. A few service user participants highlighted that they felt particularly anxious when engaging with their family, who were not supportive or understanding about their mental illness. One described how they felt that the type of support offered by their family members made it difficult to gain a sense of independence:

*"They're supportive but for a mental health sufferer… it's the support that the mental health sufferer doesn't need shall we say – like the smothering type of support is just not for me. It may be for some people, but I wanted to become independent. […] All these ideas of getting a driver's licence and all this, all went out of the window because of my mental health. And I knew that I wanted to become independent, and the smothering type of help didn't help."*

(Service user participant 14, 27-year-old male)

One service user participant of Black ethnicity noted an additional challenge imposed by the cultural background of their family and how this influenced their perception of and response to their mental illness:

*"It's an African thing, isn't it? They think it's either devilish or you're just being lazy. They don't understand. […] It makes you feel bad, 'cos you know you've got the illness and they're saying it's the devil, so that doesn't exactly make you feel good, does it?"*

(Service user participant 9, 46-year-old male)

Many service user participants discussed the positive role that their friendships played in their lives, and how this was a key source of feeling socially included. They noted in particular how their friends were a source of inspiration and accountability to maintain active social lives:

*"They inspire me, and they support me, and they encourage me, and they help me. And also, if I didn't have my friends I wouldn't have social engagements, you know? To go out, to force me to go out of the house, force me to get dressed and you know, go and meet them."*

(Service user participant 3, 38-year-old female)

Friendships were also sometimes problematic. For example, some participants described having friends who often drank alcohol and/or engaged in substance use, causing them to feel pressured to engage in similar behaviours. Others described having certain friends that were abusive towards them, noting that that these were difficult relationships to break from:

*"They're not the type of friends that I want to [be] friends with. Like [they] are quite violent and stuff, but 'cos I've known them for such a long time I feel like I'm kind of so used to how they go on."*

(Service user participant 15, 26-year-old female)

Service user participants also discussed the challenge of making new friendships. Frequently cited reasons for this were struggling to find common interests with other people and difficulty finding opportunities to meet new people. Some service users noted that many of the people they met and mixed with on a day-to-day basis were other people with mental health problems. Some felt that this was a disadvantage of attending community-based activities designed specifically for people with mental health problems:

*"I don't wanna go to these day centres where there is other [mental health] sufferers 'cos it's not the people I wanna be with; I wanna be with the people I decide to be with. […] It's not easy to like…the only people you can meet is another sufferer like you."*

(Service user participant 7, 41-year-old female)

Many service users felt that this kind of social environment, where the only thing they had in common with other people was having mental ill health, reminded them of their illness, and made them feel as if their diagnosis was the main thing that defined them. Others, however, described feeling more at ease socially interacting with and relating to other people with mental health problems, who had had similar experiences to their own:

*"I'm quite happy meeting people who've got some element of a mental illness because there's that element of relations straight away. […] They're gonna get it to a certain extent, you know, whereas regular people are not."*

(Service user participant 14, 27-year-old male)

Other concerns expressed in relation to social interactions and friendships included an inability to find things in common with people outside a mental health context and difficulty in trusting people, often based on previous negative experiences in relationships, which made it challenging for them to make new friends:

*"Once I get past that stage of learning to trust somebody, it's fine. It's the initial barrier of working people out; whether I am willing to invest my trust in that person or not invest my trust in that person. 'Cause I have difficulties in that area of being secure in friendships or other relationships. So then it takes a bit more effort I think to open up."*

(Service user participant 6, 31-year-old male)

Some service user participants gave examples of feeling vulnerable from others and described how this negatively impacted on their confidence and willingness to improve their social inclusion. Several described having had abusive relationships, which they found difficult to leave. They described how these negative experiences made them more likely to socially withdraw:

*"I've been getting texts […] like telling me to go and kill myself and stuff like that. And so that kind of makes me feel a bit like I don't really wanna go [to an activity] because I don't wanna see her."*

(Service user participant 15, 26-year-old female)

Others described living in areas where crime was a common occurrence, which also made them nervous to go out and socialise:

*"I get very nervous and I'm also very scared of being attacked, and stuff like that. […] I think [the last time I was attacked] was when I was living in the shared house, and I walked back one night. I'd had a few drinks and these… about five guys attacked me and punched me a few times – for no reason."*

(Service user participant 13, 36-year-old male)

## System of mental healthcare and support

Participants discussed the role of the mental health care and support system, and how this played an integral role in their social inclusion. Service user participants highlighted the importance of having a positive, non-judgemental relationship with the staff members they engaged with. Service users who had this kind of supportive relationship with their care

coordinators and/or key workers described how this encouraged them to handle practical tasks and connect with other services:

*"I think with my care coordinator my relationship's really good; it's taken a long time to build it, but it's working well, yeah. Which is helping me, because now I'm more able to connect with other services, with help as well. [They help with] organising stuff I think, mainly, and just listening is probably the biggest thing. Just listening when I talk, and not judging or criticising."*

(Service user participant 11, 32-year-old male)

Staff participants also felt that social inclusion was facilitated by getting to know each service user individually and being non-judgemental when offering them support for getting involved in their community:

*"It's getting to know people and getting to know their interests and then trying to find things that might be suitable. But also kind of this acceptance of if they don't manage it, it's fine as well, and just kind of a non-judgmental way of working with it."*

(Staff focus group 1)

By contrast, a few service user participants described how non-supportive relationships with members of their care team had negatively impacted their self-confidence and limited their attempts at social inclusion:

*"I had one CPN, Community Psychiatric Nurse, who… I came out of hospital, and I was very unwell, obviously, and she said to me 'I don't think you'll ever be able to work again.' [It was like] 'I don't think you'll ever be able to do anything, you know, of value again.'"*

(Service user participant 3, 38-year-old female)

Staff knowledge about community-based activities and resources that were available and suitable for the service users they worked with was highlighted as a key facilitator for promoting social inclusion by both service user and staff participants. Some service user participants felt that the opportunities and resources available to them were typically not well advertised in the services they attended. Others noted that the key factor, when they did try new activities, was when a staff member was well informed about what was available and discussed this with them:

*"It was my old inclusion worker who put me on to [music lessons]. I didn't know it was going on. You know, she knew me and she said, 'why don't you go down there?' So I said okay, I'd check it out. And now I'm there all the time."*

(Service user participant 9, 46-year-old male)

Staff participants noted the challenge of keeping up to date on what was on offer in the community, but felt that this was a key aspect of offering individualised support:

*"If somebody was only seeing me and I was only on their team for a few months […] I, with the best intentions, just might not know that there was some brand new service for young people who are interested in music, or whatever. […] I felt like there probably was more around that I would only find out about and then realise 'oh gosh that guy I saw three months ago would have been great for this.'"*

(Staff focus group 3)

A related issue that was often discussed by participants were the pressures and demands on the mental healthcare system and how certain aspects of social inclusion therefore ended up being under-prioritised. Some service user participants

noted that while they typically had valuable support with practical issues, such as budgeting, filling out administrative forms, and housing, it was much more difficult to access support for activities that they wanted to try based on their interests rather than necessity. They described the considerable negative impact this had on them and their outlook on life:

*"There is no more extra money coming in to do extra activities, so you are confined to the… you don't live a life. You are surviving. You're just glad to be alive, which is very different to being alive and having a life."*

(Service user participant 7, 41-year-old female)

Staff participants echoed this view, noting that strict budgeting, increased caseloads, and staffing shortages were barriers to their capacity to offer more than just fundamental aspects of support:

*"In terms of how I'm currently working, I think I've got less time to think about expanding that area [of social inclusion] for somebody now, because we're under so much pressure – our caseloads are climbing, we're having to do so many more of the basic things with so many more people."*

(Staff focus group 4)

## Finances and access to opportunities

Most service user participants had considerable financial worries, which acted as a barrier to social inclusion. The challenges of not being able to consistently work or work at all due to their mental health left many in poverty. Some discussed having been homeless, and how not having an address made it difficult for them to access relevant services and mental health support:

*"If you haven't got a permanent address, you can't really get a GP for instance. It's all just A&E [accident & emergency] and stuff. And A&E are not really suitable for long-term conditions."*

(Service user participant 11, 32-year-old male)

Others described how the financial support they did receive from government benefits allowed them to survive in a basic sense, but that it did not give them financial security to pursue the same goals as other members of society. This then enhanced their feelings of isolation and separateness.

*"I would love to go on a holiday. […] I would like to buy a car; I would like to buy my little house and to have a mortgage and be worried about the mortgage. I will not have all of this; I will never have all of this. And it's in this society where I live – you know, we westerners – that's what you do, you do a career, then you meet someone, you get maybe things together, and I don't have this. All I do is take the medicine."*

(Service user participant 7, 41-year-old female)

Staff participants echoed this, noting that if service users could pay for the basic necessities, this was often seen as all the financial support they might need:

*"I think the amount of care plan meetings that I've done, and it's been like you know, sort of glossed over in terms of 'are your benefits all in place' – that kind of thing. 'Yes, yes, yes, finances are fine', but actually if that person feels poor and feels the burden of not having that money, it's a completely different thing."*

(Staff focus group 2)

Staff participants also discussed how a person's immigration status could make social inclusion more challenging. They noted that asylum seekers who suffer from mental health problems often struggled to access services or to qualify for benefits, severely hindering their ability to integrate in society:

*"For me, the biggest group of people who suffer social exclusion are people who are maybe illegal immigrants. […] They haven't worked here long enough to pay National Insurance and tax and qualify for benefits, because they [don't] meet the habitual residency tests."*

(Staff focus group 4)

Many service user participants discussed the key role of employment and education, how not being able to access these opportunities often played a central role in their social inclusion. Several mentioned the challenge of not being able to work or study due to the impact of their mental health problems:

*"You've got a double problem, because you've got your illness to deal with and then you've got to try and study the thing. So, the people who don't have an illness they find it difficult enough, but if you've got an illness and you are trying to study it's almost impossible."*

(Service user participant 9, 46-year-old male)

These participants described the personal and social challenges of this, as well as the more obvious financial challenges. Some discussed how not being able to work/study made it difficult to feel completely fulfilled in their daily lives and made it more difficult to meet with other people on a regular basis. They also noted how challenging it was to socialise and relate to other people without having a job or course to talk about:

*"If I'm not getting out of the house and I'm not working, if I do meet up with friends – well, what do I speak about?"*

(Service user participant 10, 35-year-old female)

Other participants who did have education or employment opportunities also described the beneficial impact this had on their social inclusion and mental health in general. Some noted that they found these roles to be rewarding, both personally and financially, while others highlighted that they were an effective way to focus on something other than their mental health:

*"It's very rewarding work; it certainly helped me develop. I've got more confidence now. […] [I'm] more well-rounded. [...] I can't tell you how much that's helped me."*

(Service user participant 2, 58-year-old female)

**Stigma**

Stigma was a key theme that was present throughout the interviews. Service user participants were affected by stigma in multiple ways, all of which negatively impacted their social inclusion. One example of this was self-stigma, which involved participants' feelings of shame about their mental illness and how they felt it limited them. One participant described how receiving their diagnosis initially evoked strongly negative feelings about themselves and what it meant to be mentally ill:

*"When I got told I was schizophrenic I was like, 'does that mean I'm evil?' You know, I was like… I couldn't believe it, I thought because I didn't really know anything about it, I just thought it was a really bad thing. […] I thought it was like a psychopath."*

(Service user participant 13, 36-year-old male)

Some service user participants described the impact that their own internalised stigma had on their social inclusion. The anticipation that individuals outside of a mental health context would behave in a discriminatory or exclusionary way towards them led to some preferring to avoid such potentially distressing interactions:

*"I am a bit worried stepping out of that mental health umbrella – even the meetups I go to, they're all mental health related. I think I exclude myself rather than… well, rather than giving other people the chance to exclude me, if that makes sense."*

(Service user participant 14, 27-year-old male)

Service user participants also described how previous experiences of stigma had a detrimental effect on their willingness to get involved in various aspects of society. Many discussed occasions where they received a strongly negative response from family, friends, and romantic partners after disclosing their mental illness to them:

*"You just don't have friends. All the people that you've met, you know you tell them the truth about your illness – you don't see them again. So yeah, there's definitely a, you know, certain behaviour to all the mentally ill people. You get excluded."*

(Service user participant 9, 46-year-old male)

These experiences of discrimination from important people in their lives was a further reason many participants felt reluctant to pursue social relationships and activities. Others described experiences of being stigmatised in mental health treatment settings, which made it difficult to trust mental health professionals. They noted that there were some diagnoses which seemed to be more susceptible to stigma in mental health contexts than others, including schizophrenia and personality disorders:

*"I think out of all the illnesses personality disorder is stigmatised the most by psychiatrists, because we can come across quite… quite difficult. Lots of people tend to avoid us, which is probably the last thing we'd want."*

(Service user participant 11, 32-year-old male)

Staff participants echoed this view, noting that certain diagnoses or severity of mental illness were more discriminated against than others in professional and wider contexts.

*"I wonder whether there's a bit of a hierarchy as well of perhaps conditions, so someone who's got say anxiety or depression is perhaps a little bit less stigmatised these days, whereas people I think are still a bit more frightened of like the say psychotic illnesses and would be less, perhaps willing to include people if they knew that they had those conditions."*

(Staff focus group 3)

Service user participants discussed how this stigma also extended to professional and educational contexts. Many highlighted how they felt pressure to hide their diagnosis and related symptoms from their employers, with many citing negative experiences when they had disclosed this in the past:

*"I've been to work when I have been sick – there's a lot of stigma around. So if you get ill while you are in the middle of a course or while you are teaching, you end up, if you go back there, being very stigmatised or people not even talking to you at all."*

(Service user participant 2, 58-year-old female)

In some cases, service users had encountered negative responses from teachers and/or supervisors, including an ignorance of and lack of sensitivity towards symptoms of mental illness and how this affected them. Such experiences made them reluctant to seek out employment and education opportunities.

Some participants with psychosis noted that many people were afraid of individuals with this diagnosis, owing to a lack of understanding about the condition. They highlighted a particular misconception that individuals with psychosis were exceptionally aggressive or violent:

*"I think people [generally] are quite paranoid. [...] They think that people [with psychosis] could, you know, get violent."*

(Service user participant 4, 32-year-old male)

They described how this stigma can often lead to individuals excluding them from social activities or generally avoiding them. Others noted situations where people they were with had made crude comments or jokes about mental illnesses:

*"If someone was to make a joke about mental health issues, it makes me think 'oh I've got mental health issues, so I'm inferior as well'. So that's… that's an issue."*

(Service user participant 6, 31-year-old male)

Staff participants echoed this, agreeing that societal stigma acted as a considerable barrier to participants' social inclusion and the ways that they attempted to help participants with this:

*"I think there's only so much we can do if society and the general public still discriminate against the people that we're trying to include. We can't force them to be more widely accepted than they are. [...] I think until society really changes its opinion of mental health, there's always gonna be social exclusion of these individuals."*

(Staff focus group 2)

In general, the majority of staff and service user participants felt that it was necessary to ignore stigma in order to pursue social inclusion, as this was seen as something that was unavoidable for them across many different contexts. Some felt that a lot of stigma arose from people's lack of understanding about mental health problems, and that they felt that talking about their own struggles with other people was an important way to help them better understand these issues.

## Discussion

### Main findings

Our findings show that social inclusion among people with severe mental illness is a complex concept that can be influenced by factors at all levels of society. Results concur with the theoretical drivers of social exclusion as described by Boardman et al. [2], in that barriers and facilitators to social inclusion can operate at macro, meso and micro levels [25,26]. At the micro level, social inclusion can be influenced by an individual's mental health, beliefs, internalised stigma, self-confidence, and coping styles. At the meso level, influences include an individual's social network, their mental health

care team, and community resources. At the macro level, factors such as the wider health and social care system, socio-economic and political factors, and societal attitudes can act as social inclusion barriers and facilitators.

Findings suggest that social, familial, and romantic relationships for this group can act as both barriers and facilitators to their social inclusion, depending on whether support, understanding, and encouragement were absent or present in these relationships. This indicates the complexity of this need and emphasises that social contact alone is not sufficient to help promote social inclusion, but that the quality of these relationships and whether they provide a sense of belonging is crucial. This is consistent with Social Identity Theory, which suggests that when personal relationships provide stability, meaning, purpose, and direction, this can have a positive impact on mental health and act as a buffer from negative circumstances [30,31]. However, forming a sense of belonging with unhealthy, unsupportive, and/or heavily stigmatised groups can act as a source of stress that is detrimental to wellbeing [30,32]. These points, and the present findings, highlight the value of promoting interventions and policy decisions that consider the social dimension of health and can capitalise on the potential advantage that positive relationships can offer to health and wellbeing [30].

Relatedly, our findings highlight the conflict between supporting people with mental illness to live as independently as possible, but acknowledging that this may increase the vulnerability of this group to experience harm, violence, and/or discrimination from others. The participants in our study noted that such experiences made them more likely to socially withdraw, negatively impacting on their social inclusion and general wellbeing. This point has similarly been highlighted in a study of people with severe mental illness living in supported accommodation, whereby people in less supported mental health supported accommodation were more likely to be victims of crime or exploitation than those in housing with higher levels of support [33]. This emphasises the importance of individualised care to understand each person's own circumstances and preferences in line with recovery-oriented practice, the fundamentals of which promote service users' autonomy, right to informed choice, and dignity of risk [28].

The number, complexity, and range of the barriers and facilitators identified is consistent with existing conceptualisations of social inclusion as a multidimensional and layered construct [1,2,24] and emphasises the need for a sophisticated and multi-faceted approach to intervention. Solutions are needed to target key barriers at all levels, for example: improving mental health and wellbeing at the micro level; training for mental healthcare staff on principles of validation, autonomy, and respect, and promoting knowledge of community resources at the meso level; and improving societal attitudes towards and understanding of mental illness at the macro level.

### Implications for research

There are several important research implications related to these findings. One key research priority is to develop effective ways to help people with mental illness meet their currently often unmet needs for friendships and romantic relationships. Several recent interventions have been developed with the aim of promoting social connections and reducing loneliness in people with severe mental illness, including the *Community Navigator* programme [34], the *SCENE* programme [35], and the *Connecting People* intervention [36]. However, the effectiveness of such interventions promoting social connections and improving mental health in this population remains to be established, constituting an important research priority.

The study findings reinforce the key issue of stigma in mental health, and the multiple ways that this can impact on service users' social inclusion. Importantly, stigma acted as a social inclusion barrier in almost every aspect of participants' life, including their own self-perception. Many participants referred to themselves as 'sufferers', which may be an additional indicator of self-stigma. Stigma and discrimination are associated with reduced help-seeking [37], worse clinical and personal recovery outcomes [38], increased loneliness [39], and reduced quality of life [40] for people with mental illness. This emphasises the pressing need for effective anti-discrimination and social cohesion programmes to target and manage stigma towards individuals with severe mental illness at an individual and societal level [41]. Research demonstrates that both targeted and population-level anti-stigma campaigns can produce small to moderate

effects and improved attitudes towards people with mental illness [42–44]. However, continued research is needed to understand how best to promote long-term reduction of discrimination, prejudice, and exclusion of people with mental health problems [43].

Poverty and the impact of the welfare benefits system were also highlighted as key barriers to social inclusion for this population. Importantly, it has been demonstrated that lower levels of benefits and greater conditionality of social security payments are associated with mental health harms [45]. In England and internationally, personal budgets – i.e., state-funded direct payments to individuals to spend on care and support, often provided by informal carers – offer a possible mechanism to address financial barriers to social inclusion, but more robust evaluation of their effect is required [46]. This underlines the importance of developing and evaluating programmes to help with debt and financial management at an individual level, and an audit of mental health impacts of policy relating to the welfare benefits system at a governmental level.

While a wide range of interventions relevant for social inclusion related outcomes have been developed for people with mental health conditions, the current evidence base is not large and the most effective types of care and support remain to be established in many areas [47,48]. This represents an important research priority.

## Implications for practice

These findings reinforce the need for a biopsychosocial approach to mental health services, to promote social inclusion among people with severe mental illness. Biological approaches include medication to alleviate mental health symptoms without causing stigmatising side-effects such as weight gain and fatigue. Psychological approaches should address lack of confidence, low self-esteem, internalised stigma, problematic coping behaviours, and heightened perceptions of social threat. Social approaches include those that promote positive social support, employment/education opportunities, and community participation.

Community-based social groups and opportunities were identified as valuable facilitators to social inclusion among both staff and service users in this study. However, service users noted that they did not feel well-informed about such opportunities and were often dependent on how knowledgeable staff were about these activities, while staff noted the considerable challenge of maintaining up-do-date awareness of what options were available. This highlights the value of specialist knowledge about these local community opportunities, and suggests a need for delineated staff roles within mental health teams that specialise in identifying such activities and supporting service users to access them. Such staff roles may include social prescribers or community navigators, who offer this expertise without having considerable additional demands on their time, such as mental health care and/or support responsibilities. Additionally, voluntary sector services providing mental health social care are valued by many people with mental health problems for being rooted within local communities, and are perceived by some as easier to trust and more inclusive than NHS mental health services [49]. Understanding the role of non-health service organisations in supporting social inclusion for people with mental health problems is also of high interest.

Finally, our findings demonstrate the variation of service users and staff views on social inclusion and the different needs, preferences, and priorities of people with severe mental illness. These may include key intersectional factors, such as age, ethnic background, and poverty, which can potentially exacerbate challenges to social inclusion in this population [2]. Such findings emphasise that there is no one-size fits all solution for promoting social inclusion in this population. They also highlight the need for assessment tools that allow for an individualised approach to care, including patient-reported outcome measures, which can capture service users' own perspectives on goal attainment, quality of life, and social inclusion [50]. Examples of such measures which may be useful in practice include the SInQUE questionnaire, a patient-reported outcome measure to assess social inclusion in individuals with severe mental illness [4,22,51]. This individualised approach to care is in line with World Health Organisation recommendations for recovery-based practice in mental health care provision [52].

## Strengths and limitations

It is important to reflect on various strengths and limitations of the current study. Service user participants were purposively sampled from individuals who participated a larger study to develop a measure of social inclusion [4]. The response rate to this original study was excellent, promoting the likelihood that a range of views from mental health service users within the two study boroughs were reflected. The inclusion of service users with different mental health diagnoses and of staff working in a variety of mental health roles is also a strength of the study, as the study findings are based on a variety of experiences. However, this does limit our understanding of the extent to which these findings may relate to any specific diagnostic group or service setting. Furthermore, staff and service users were recruited from two NHS trusts in London. Therefore, important nuances associated with different geographical locations or social circumstances that have important relevance for social inclusion, such as living in a rural setting, are unlikely to be captured in the current findings. Our interviews also reflect the views of staff working in health services, rather than staff working in Local Authority or voluntary sector mental health social care roles.

Although service user participants were well balanced in terms of gender, the majority were of White ethnicity. This may mean that important insights that are particularly relevant for mental health service users from ethnic minorities were missed. This is especially important given that people from ethnic minorities are more likely to be diagnosed with a severe mental illness [53] and are more likely to experience inequalities in pathways to psychiatric care when compared to individuals from White ethnic backgrounds [54].

It is important to note that these interviews were conducted between 2016–2017, before the COVID-19 pandemic. Therefore, important nuances relating to social exclusion that were related to this period are not represented in this study. Finally, the original interviewers were not involved in the analysis process. Therefore, results have not benefitted from any additional insights gained through direct experience of interviewing participants.

## Conclusion

Social inclusion, encompassing aspects of social and individual recovery, constitutes a highly valued goal for people with severe mental illness [27]. Furthermore, Connectedness, Identity, Meaning, and Empowerment, which may ensue from social inclusion, are key aspects of the CHIME framework for personal recovery in mental health [55] and helping service users achieve these aims should be a current research priority [56]. This study highlights several key areas to take research and practice relating to social exclusion forward, contributing to the wider goal of supporting better social outcomes and quality of life for people with severe mental illness.

## Supporting information

**S1 File. Interview topic guides.**
(DOCX)

## Acknowledgments

We would like to thank service users and staff from the South West London and St George's Mental Health Trust and the Camden and Islington NHS Foundation Trust for their participation and support with recruitment. We would also like to thank the members of the Advisory Group, the Peer Expertise in Education and Research (PEER) group at St George's University London, and the Service User Research Forum (SURF) at University College London for their feedback on the topic guides for this study.

## Author contributions

**Conceptualization:** Brynmor Lloyd-Evans, Helen Killaspy, Gillian Mezey.

**Formal analysis:** Sharon Eager.

**Funding acquisition:** Gillian Mezey.

**Investigation:** Jennifer Bousfield, Isobel Harrison.

**Methodology:** Brynmor Lloyd-Evans, Helen Killaspy, Gillian Mezey.

**Project administration:** Brynmor Lloyd-Evans, Jennifer Bousfield, Isobel Harrison, Helen Killaspy, Gillian Mezey.

**Supervision:** Brynmor Lloyd-Evans, Helen Killaspy, Gillian Mezey.

**Validation:** Brynmor Lloyd-Evans, Joanna C, Megan Downey, Helen Killaspy, Gillian Mezey.

**Visualization:** Sharon Eager, Brynmor Lloyd-Evans.

**Writing – original draft:** Sharon Eager.

**Writing – review & editing:** Sharon Eager, Brynmor Lloyd-Evans, Jennifer Bousfield, Joanna C, Megan Downey, Isobel Harrison, Helen Killaspy, Gillian Mezey.

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
