## [Decision Letter · Decision Letter 0]

30 Sep 2024

PMEN-D-24-00355

Barriers and facilitators to social inclusion among people with severe mental illness: a qualitative study.

PLOS Mental Health

Dear Dr. Eager,

Thank you for submitting your manuscript to PLOS Mental Health. After careful consideration, we feel that it has merit but does not fully meet PLOS Mental Health’s publication criteria as it currently stands. Therefore, we invite you to submit a revised version of the manuscript that addresses the points raised during the review process.

We look forward to receiving your revised manuscript.

Kind regards,

Sauharda Rai, Ph.D

Academic Editor

PLOS Mental Health

Journal Requirements:

**Please only choose the relevant sentences from below**

1. Please clarify all sources of funding (financial or material support) for your study. List the grants (with grant number) or organizations (with url) that supported your study, including funding received from your institution. 

2. State the initials, alongside each funding source, of each author to receive each grant.

3. State what role the funders took in the study. If the funders had no role in your study, please state: “The funders had no role in study design, data collection and analysis, decision to publish, or preparation of the manuscript.”

4. If any authors received a salary from any of your funders, please state which authors and which funders.

Additional Editor Comments (if provided):

Interesting article and great to see perception of people with lived experience included. In addition to the reviewer's comment, the authors are requested to address the following points:

1. Methods: Diverse and inclusive effort has been taken for coding framework development but more clarity is required in terms of actual coding of the interviews. Did the other authors (line 153-155) code only one common interview/FGD or was there different interview/FGD each? If it was the former, did the authors conduct Inter-rater reliability (IRR). Please report that than just mentioning "high level of agreement was observed"

2. Methods : The first author coded all the interviews and FGDs. Were the interviews used for code development coded again? Can the authors add if there were any efforts to get multiple perspective during the analysis phase.

3. Ethics: Based on what the authors report on the result section, there were service users with suicidal ideation. Please clarify how the team handled these situations. Was there a referral service in place for such identified cases?

4. Results: If there are more details on the demographic details of the service users (Table 1), consider adding that. E.g. how many years since diagnosis and in treatment. Were they all taking medication, counseling or both? This additional details would help to situate the findings.

5. Results: Table 2. Additional demographic details would be helpful. e.g. gender, years of experience, ethnicity

6. In line with what other reviewers mentioned, please restructure the result section for better readability. Table 3 can be revised based on how you present your discussion across three levels. Is there a framework across these themes can be developed through your findings describing pathways of mental illness and social inclusion.

Reviewers' comments:

Reviewer's Responses to Questions

**Comments to the Author**

1. Does this manuscript meet PLOS Mental Health’s publication criteria ? Is the manuscript technically sound, and do the data support the conclusions? The manuscript must describe methodologically and ethically rigorous research with conclusions that are appropriately drawn based on the data presented.

Reviewer #1: Yes

Reviewer #2: Yes

Reviewer #3: Yes

Reviewer #4: Yes

2. Has the statistical analysis been performed appropriately and rigorously?

Reviewer #1: N/A

Reviewer #2: N/A

Reviewer #3: Yes

Reviewer #4: Yes

3. Have the authors made all data underlying the findings in their manuscript fully available (please refer to the Data Availability Statement at the start of the manuscript PDF file)?

Reviewer #1: Yes

Reviewer #2: No

Reviewer #3: No

Reviewer #4: No

4. Is the manuscript presented in an intelligible fashion and written in standard English?

Reviewer #1: Yes

Reviewer #2: Yes

Reviewer #3: Yes

Reviewer #4: Yes

5. Review Comments to the Author

Reviewer #1: This is an important article outlining the barriers and facilitators for social inclusion for people living with mental illness. The study presents original research with qualitative findings from another study. The qualitative analysis and conclusions are consistent with data reporting for qualitative research and therefore meets the criteria of research integrity and credibility. Of particular relevance is attention to both micro, meso and macro level factors contributing to service user experiences.

One area that is not as clear is the section on limitations and strengths which suggest that bias of reporting may have influenced the findings. However, in critical qualitative research such as reflexive thematic analysis this is not necessarity the case because you would have "controlled" for this given the multiple perspectives and heterogeniety of the samples. Also the mental health providers are experts because they know these experiences intimately as do the service user participants. The recommendation then maybe to re consider the word "bias" and replace this with ways that the participants were relexive about their participation? Please also consider providing a reference for thematic analysis in Data analysis section (line 148). I was unable to see a reference for thematic analysis? perhaps I missed this. If using Bruan and Clarke 2022- Thematic Analysis a practical guide suggests that anlysis is interpretive where roles of the researcher is value laden not external to meaning making.

The second area of great importance is the discussion about ethnic minorities more likely to be diagnosed with mental illness - there is one reference for this and is this from the UK? There was a study done in Canada that did not find this to be the case: Rotenberg M, Tuck A, McKenzie K. The role of ethnicity in involuntary psychiatric admission in Toronto, Canada in clients presenting with psychosis. Psychosis. 2019 Jul 3;11(3):273–6. DOI: 10.1080/17522439.2019.1612461 /other studies also suggest that ethnic minority groups that are racialized are less likely to access mental health services due to stigma, social isolation and belonging generally, and specifically during COVID-19 pandemic. This point may require futher articulation and or reflection to unpack the nuance and contradictions.

Overall this article reads well and adds an important body of knowledge to research implications and practice, particularly on barriers and facilitators for inclusion and belonging for people living with mental illness and who carry stigmitizing diagnosis.

Thank you for the opportunity to read this important work!

Reviewer #2: Thank you for the opportunity to review this interesting and well-written paper. I appreciate that the authors have included perspectives from both individuals with lived experience and mental health professionals.

Methods

• On page 5, line 87, I recommend including the definition of severe mental illness much earlier in the paper (in the introduction). Additionally, please provide a reference for this definition to clarify how the term was operationalized and how you arrived at this specific definition.

• On page 5, line 90, it would strengthen the paper to include an argument for why interviews were conducted with service users while focus groups were held with staff. Please explain the rationale behind choosing these specific methods for each group, with references.

• Furthermore, it would be helpful to address why staff participants were not remunerated for their participation. This could add important context to the study's design and ethical considerations.

Results

• Please provide an explanation for the varying numbers of participants in the focus groups. For instance, can a group of only three individuals be considered a focus group? Clarifying this point will aid in understanding the data collection process.

• I suggest mentioning somewhere in the results or methods sections any steps taken to address situations involving suicidal participants. This is crucial given the nature of the study population, and how someone mentioned this specifically in the results. Where there any steps taken to give this person support? This is important in highlighting ethics in undertaking research with “at-risk” populations.

• The example regarding cultural backgrounds is interesting. It would be valuable to expand on this section and in the discussion as well and provide additional examples to give a more comprehensive view of the cultural factors at play.

• The concept of "sufferer" is being mentioned several times by the participants. It will be interesting to unpack this where you talk about (internal) stigma in the results and discussion. Consider elaborating on this term and how it is perceived or used within the context of the study, and how it re-enforces self-stigma.

• On page 24, line 434, the term "A&E" is used. For clarity, please define this abbreviation for an international audience who may not be familiar with the term.

Discussion

• In the discussion section, you touch upon perceptions at the organisational level. However, these aspects are not reflected in the results. Were there any findings related to organizational perceptions that could be integrated into the results section? If so, including them would provide a more comprehensive discussion of the topic.

• A discussion of these organisational level barriers and facilitators more in-depth in the discussion can be added as well to strengthen the paper and give more credibility to the ecological perspective take that the authors suggest in the beginning of the results.

• The ecological perspective take can also be integrated more in the introduction. For example, from what is known in the literature, have factors at these different levels been researched? Which levels have been researched more? Is there a gap in a specific level? Do the factors interact across levels?

Reviewer #3: The manuscript investigates the barriers and facilitators to social inclusion among individuals with severe mental illness (SMI) through qualitative research. The study employs individual interviews with service users and focus groups with mental health staff, drawing participants from two NHS trusts in London. Six primary themes emerged from the inductive thematic analysis: individual characteristics, the role of mental illness, social relationships, the mental health care system, financial situation and opportunities, and stigma. I have only a few comments and suggestions:

The study includes interviews with only fifteen service users and four focus groups with sixteen staff. While this may be sufficient for qualitative exploratory research, a discussion about the sample size’s adequacy concerning the breadth of the research question would strengthen the paper.

More detailed demographic information about the service user participants would be beneficial to understand the context of the findings fully.

The process of thematic analysis should be described in more detail. For instance, how were the themes identified, and what steps were taken to ensure the reliability and validity of the coding process?

The conclusion drawn about the need for social interventions, individualized care, and stigma reduction is pertinent. However, it would be useful to provide specific recommendations or examples of potential interventions or strategies that could address these issues. This would help bridge the gap between research and practice.

Reviewer #4: Overall: This is a robust study that highlights barriers and facilitators of social inclusion in people with mental ill health. The authors have conducted a qualitative study that highlights the multi-faceted issues that impact on people’s experiences from the perspective of key stakeholders consisting of service users and staff. With some amendments and refining, this could add to the literature on social inclusion in mental health. the best ways to help with this need are not well understood

General comments:

1. There appears to be some conflation between social inclusion and exclusion in the sense that it’s not clear if they are used interchangeably or for specific points? It would be good to clarify this use with a definition and/or stick to use of the one.

2. The format of the results section is a bit hard to read could the quotes be structured in a way they are separate from main text

3. There are a lot of subthemes in the results and these need to all be contextualised in the discussion section. At current, the discussion section is not detailed enough. There is more detail in implications but this should be higher up in the discussion

4. At the moment, it’s not clear how social inclusion and exclusion are different for different mental health problems. Literature highlights the differences in agoraphobia for example in people with psychosis and this is an important thing to include in both the discussion and implications. It should be reported on how the themes are more/less relevant for different problems.

5. Line 693: The sample had an appropriate representation of ethnic minorities, suffice to the population of the UK which is a strength. It would be good to do a sub-analysis of factors that might differ in ethnic minority groups/.

Line 23: “the best ways to help with this need are not well understood “ This line of the abstract could be worded differently to make it clearer.

Line 42: Using the term occur suggests it’s a random event – could you expand on social inclusion a bit more? Including a definition of social inclusion/exclusion would be good here.

Line 50-54: This is a very long sentence, could it be split into two or summarised?

Line 59: The first sentence talks of social

Line 60: Does productivity and social integration reduce or could a better term be used?

Line 73: why is it needed? What is the gap from previous research?

Line 79: Long sentence – check grammar too.

Line 86: Check person and sentence starter.

Line 94: Sample –

Line 116: were service users involved in the development of the topic guide?

Line 139: Is there a reason why the interviews were conducted so long ago and results reported now? Has research since then been reviewed in light of this?

Line 149: Were the interviewers involved in the transcript analysis? It looks like they were different members of the team so would be good to consider this limitation in the analysis process?

Table 1: It’s great to see the majority of the sample had diagnosis of a psychotic disorder which is a difficult population to recruit – were any sub-analyses conducted for this group to see if specific barriers and facilitators applied based on diagnosis?

Table 3 would benefit from greater detail in the caption

Line 201: it would be good to support language difficulties with service user quotes if this applied to them too? did it affect those for whom English wasn’t their first language? Important to note this because majority of the sample was White.

Line 216: ‘certain’ – who does this pertain to?

Line 307: It’s important to state if this is coming from a specific demographic – assuming this isn’t the case for everyone but the specificity is important to state if it is explicitly coming from one group. Line 304 says ‘a few’: was this explicitly non-white participants or was this also white participants?

Line 419: does this issue come under finances rather than systems of support? The other points made in this subtheme are suggestive of support from them whereas the subject of budgeting and time maybe comes under finance systems?

Line 490: was stigma more of an issue for psychosis patients? I can see this is expanded on further down in relation to healthcare but it would be good to expand on the personal relationships too

Line 567: Appreciate the micro, meso, macro but I wonder if a framework might be better to highlight the hierarchy of influence.

Line 576: I suggest the suggestions for solutions should move to the end

582: if they are both barriers and facilitators, could you expand on what turns them from facilitators to barriers? E.g. addition of or lack of xyz in individual family relationships.

Line 665: reference needed

6. PLOS authors have the option to publish the peer review history of their article (what does this mean? ). If published, this will include your full peer review and any attached files.

**Do you want your identity to be public for this peer review?** For information about this choice, including consent withdrawal, please see our Privacy Policy .

Reviewer #1: **Yes: ** Nancy Clark

Reviewer #2: No

Reviewer #3: No

Reviewer #4: No

---

## [Decision Letter · Decision Letter 1]

14 Mar 2025

Barriers and facilitators to social inclusion among people with severe mental illness: a qualitative study.

PMEN-D-24-00355R1

Dear Ms Eager,

We are pleased to inform you that your manuscript 'Barriers and facilitators to social inclusion among people with severe mental illness: a qualitative study.' has been provisionally accepted for publication in PLOS Mental Health.

Best regards,

Lily Kpobi, Ph.D.

Academic Editor

PLOS Mental Health

Reviewer Comments (if any, and for reference):

Reviewer's Responses to Questions

**Comments to the Author**

1. If the authors have adequately addressed your comments raised in a previous round of review and you feel that this manuscript is now acceptable for publication, you may indicate that here to bypass the “Comments to the Author” section, enter your conflict of interest statement in the “Confidential to Editor” section, and submit your "Accept" recommendation.

Reviewer #3: All comments have been addressed

2. Does this manuscript meet PLOS Mental Health’s publication criteria ? Is the manuscript technically sound, and do the data support the conclusions? The manuscript must describe methodologically and ethically rigorous research with conclusions that are appropriately drawn based on the data presented.

Reviewer #3: Yes

3. Has the statistical analysis been performed appropriately and rigorously?

Reviewer #3: N/A

4. Have the authors made all data underlying the findings in their manuscript fully available (please refer to the Data Availability Statement at the start of the manuscript PDF file)?

Reviewer #3: Yes

5. Is the manuscript presented in an intelligible fashion and written in standard English?

Reviewer #3: Yes

6. Review Comments to the Author

Reviewer #3: no more comments.

7. PLOS authors have the option to publish the peer review history of their article (what does this mean? ). If published, this will include your full peer review and any attached files.

**Do you want your identity to be public for this peer review?** For information about this choice, including consent withdrawal, please see our Privacy Policy .

Reviewer #3: No
